# Brace compliance process in adolescents with spinal deformities: A qualitative study

**Faezeh Ghorbani**[1], **Mohammad Kamali**[2], **Hadi Ranjbar**[3]*, **Mojtaba Kamyab**[4], **Hiva Razavi**[5], **Taher Babaee**[1]

1 Department of Orthotics and Prosthetics, Rehabilitation Research Center, School of Rehabilitation Sciences, Iran University of Medical Sciences, Tehran, Iran, 2 Department of Rehabilitation Basic Sciences, Rehabilitation Research Center, School of Rehabilitation Sciences, Iran University of Medical Sciences, Tehran, Iran, 3 Mental Health Research Center, Psychosocial Health Research Institute, Iran University of Medical Sciences, Tehran, Iran, 4 Department of Orthotics and Prosthetics, California State University Dominguez Hills, Carson, CA, United States of America, 5 Department of Biomechanics, University of Nebraska at Omaha, Omaha, NE, United States of America

* hadiranjbar@yahoo.com

## Abstract

### Background

Adolescent idiopathic scoliosis affects 2–4% of adolescents aged 10–16, while Scheuermann's kyphosis affects 0.4–10% of adolescents aged 11 to 16. Over the past 50 years, brace treatment has been recommended as the most common non-surgical intervention for treating these spinal deformities. The effectiveness of brace treatment depends on the duration of brace wearing. This study aimed to understand the brace compliance process for adolescents with spinal deformities through a qualitative approach.

### Method

This study applied multicenter exploratory qualitative research with an interpretative framework and enlisted the participation of as many individuals as possible involved in brace-wearing in adolescents with spinal deformities. Semi-structured, in-depth, and face-to-face interviews and telephone conversations from September 2020 to May 2021 were conducted. The recorded audio of each interview was typed into Word software with each personal code. The content analysis method was used to analyze the data.

### Results

Seventy-four participants were interviewed, including 32 adolescents treated with braces and their parents (27 mothers, five fathers), six orthotists, two physiotherapists, and two spine surgeons. Following data analysis, four main categories, 14 categories, and 69 subcategories of 2403 related codes were discovered.

### Conclusion

Based on the analysis of the current qualitative research, adolescents with spinal deformities experience extensive challenges in the treatment process, which can affect the results

**Data Availability Statement:** All relevant data are within the manuscript and its Supporting Information files except the participant's names

**Funding:** The author(s) received no specific funding for this work.

**Competing interests:** The authors have declared that no competing interests exist.

and brace intervention efficacy. The current research findings showed that every adolescent goes through similar but unique conditions during the treatment. The importance of considering each adolescent's specific conditions and characteristics and providing functional solutions and support was understood to help them navigate critical situations more quickly and achieve effective treatment outcomes.

## Introduction

Adolescent idiopathic scoliosis (AIS) and Scheuermann's kyphosis (SK) are the most common spinal deformities during adolescence. The most frequently mentioned reason why adolescents with spinal deformities seek therapy at orthopedic clinics is dissatisfaction with their appearance [1, 2].

Exercise therapy, braces, and surgery are commonly prescribed for immature adolescents with spinal deformities, which are determined depending on the severity of the curve. In conjunction with exercise therapy, bracing is an effective non-surgical treatment for AIS [3] and SK [4]. Brace compliance is critical for treatment effectiveness [5].

Brace compliance may be quite laborious due to the significant changes in adolescents' appearance with a brace. Moreover, adolescents are in a formative growth stage, exhibit greater mood sensitivity to the environment, and resist change [6, 7]. Due to these difficulties and some unknown reasons impacting brace compliance, researchers discovered that nearly half of adolescents with spinal deformities do not follow treatment instructions and stop wearing the braces [8, 9].

Multiple factors, including physical comfort and social support, influence brace compliance [5, 10, 11]. As a result, compliance is a dynamic process impacted and constrained by various variables [8, 9]. This barrier is time-consuming, and completing the compliance procedure from the start to the end of brace treatment is critical for optimal outcomes. Therefore, comprehending the brace compliance process can help clinicians identify distinct facets of this issue and retrieve buried information. This way, the effective factors on brace compliance could be better understood.

Studies exploring living day-to-day with AIS and SK are rare. Qualitative description is a research method used in healthcare to gain insights from participants with direct experience with a phenomenon. This method is particularly useful when little is known about the participants' experiences and can inform healthcare interventions [12]. Hence, this study aims to explore the brace compliance process in adolescents with spinal deformities.

## Methods

### Study design

The present study utilized a qualitative descriptive design with an interpretive framework [13]. The data were gathered through face-to-face, in-depth, and semi-structured interviews [14] and phone conversations and analyzed using content analysis. The Iran University of Medical Sciences ethics committee approved the study protocol (NO.IR.IUMS.REC.1399.428).

### Research team

Before conducting the interviews, the study's authors and experts in health-related research held three face-to-face meetings to explain the study theory framework and guide interview

questions. The interviews were conducted using a set of guide questions devised by the research team (Table 1).

The interviews were carried out by the first author (F.G.), who had five years of experience treating adolescents with spinal deformities. She was a Ph.D. candidate who attended several qualitative research and interviewing workshops.

## Recruitment

Participants were recruited from three orthotic and prosthetic centers in Tehran that manufactured spinal braces. Potential participants were selected based on predefined criteria and

**Table 1. Guide questions during the interviews.**

| | Questions from the adolescents |
|---|---|
| 1 | How did you first notice a deformity in your spine? How did you feel? |
| 2 | When did you first see the brace? What did you think about it? |
| 3 | How did you feel when you realized you needed to wear the brace? How long did it take for this feeling to go away? |
| 4 | How was the first time wearing the brace? |
| 5 | How is your day from the morning to night? |
| 6 | How often do you wear your brace? Why do you not wear your brace sometimes? |
| 7 | What were your parents' reactions to you wearing the brace? What do they think about the brace? |
| 8 | Is there anyone among your friends/ classmates whose opinion is important to you? How was your friend's reaction to you wearing the brace? |
| 9 | What is that thing if you want to compare or resemble your brace to something else? Why do you think that? |
| 10 | How different is your life after wearing the brace compared to before? What were those changes? |
| 11 | How do you feel about meeting new people? How do you respond if they ask you what is this device? How do you introduce the brace? |
| 12 | What do your mother, father, and siblings tell you about the brace? What do they think about the brace? How do they help you regarding the brace? What do they recommend for you when wearing the brace? |
| 13 | How long did it take you to get out of the initial shock? Have you accepted that you have to wear a brace? |
| 14 | Which daily activities could you not do because of the brace, and others need to help you with those activities? |
| 15 | When encountering new people, do you or your parents introduce the brace? How do your parents introduce the brace? |
| 16 | Did wearing the brace change the kinds of play you used to do? How did they change? |
| 17 | Do you or your mother clean the brace? How often do you clean the brace? |
| 18 | Have you ever decided not to wear the brace anymore? What caused that decision? What changed your mind about wearing the brace again? |
| 19 | Did anything special happen to you while you were wearing the brace? Do you have a specific memory of the brace? |
| | Questions from the parents |
| 1 | Tell me about the first time you noticed a spinal deformity in your child. How did you notice that? |
| 2 | How did you feel when others told you that your child had a spinal deformity? |
| 3 | How long did the shock you mentioned take to wear off? Did the doctor and the therapists help you in wearing the shock off? |
| 4 | How did you feel when your child first wore the brace? Tell me about the first days of brace treatment. |
| 5 | Does your child have a problem with their daily activities? What was their reaction to these problems? |
| 6 | Was there any party or gathering where your child had to wear their brace? Did you or your child introduce the brace? |
| 7 | What was your greatest challenge during this period of dealing with the brace? |
| 8 | What do you tell them to increase their motivation to wear the brace and do the exercises? |
| 9 | Have you brought your child to a therapist during the brace treatment? |
| 10 | Do you think the quarantine time made your child wear the brace more? |
| 11 | What factors encourage or discourage your child to wear the brace? |

invited to participate. Informed written parental consent and verbal adolescent assent were obtained if they agreed to participate. To ensure confidentiality, only two researchers (F.G. and H.R.) had access to the audio files.

## Selective criteria

Inclusion criteria for the study required adolescents who 1) had AIS or SK, 2) were between 10–14 years old at the time of deformity diagnosis and beginning the brace treatment, 3) had undergone brace treatment for more than three months, and 4) had no history of spinal surgery or mental disorders. Furthermore, the parents were interviewed about their children's spinal deformity experiences from the beginning of their deformity diagnosis. Interviews were also conducted with therapists experienced in spinal deformities, such as orthotists, physiotherapists, and orthopedic surgeons, to provide triangulation. The study began with convenience sampling and then used purposeful sampling to obtain further information from participants across different stages of treatment.

## Data collection

Data was collected from September 2020 to May 2021, and interview analysis proceeded simultaneously. The interviews were conducted until no further data or conceptual material was extracted and data saturation was obtained. The participants were informed about ceasing the conversation whenever they wanted. In a face-to-face session, the interviewer told the child a brief about herself (name, occupation, place of work) and explained the study's aim in child-friendly terms. Brace compliance was the main topic discussed during the interview.

Over time, to enhance openness, additional follow-up questions were asked depending on the information provided by the participants. Furthermore, the participants and their parents were requested to conduct a phone interview if there was more to say. The interviews were conducted in Persian and lasted 28–56 minutes.

## Data analysis

Data analysis was based on the Lundman and Graneheim method [15], and the recorded files were turned into written language (verbatim). Non-verbal gestures and body movements of people were mentioned in the writing verbatim. The interview transcripts were read several times to obtain a general understanding of the content and data immersion. The content analysis involved code generation from the most descriptive material, narrowing it down, and finding the code groupings (categories) [16]. Initial coding and inductively developing main categories from the codes were performed.

The interviewer initially inductively coded two interviews, compared notes, then coded two more interviews and established a coding structure agreed upon by other authors. Consequently, groups of significant codes, points, or materials were created, which enabled the formation of the titles characterizing the participants' experiences. Semantic units were identified for each interview, and key phrases were extracted. Initial codes were kept very close to the research data [17]. The concepts were generated and summarized based on the study questions and appropriate codes for each item. Then, the remaining interviews were coded, and the codes were organized into categories. The extracted original codes were summarized based on their conceptual similarities in categories through ongoing data analysis and comparison. These categories were explored as the interviews progressed and by constructing data codes and category charts. Subsequently, the research team sessions were held to combine the analysis results to represent the participants' experiences. The 32-item checklist of consolidated

criteria for reporting qualitative studies (COREQ) [18] was followed, and this checklist is attached in the supporting information file S1 Table.

## Rigor

The rigor of our findings was ensured through collaborative coding processes, adherence to established research methodologies [19], and the application of Lincoln and Guba's criteria for credibility, dependability, confirmability, and transferability to achieve trustworthiness [20]. Regular team meetings facilitated the refinement of data categories, and participant validation of findings contributed to the study's integrity [12]. Data saturation guided the determination of the sample size. Table 2 lists the techniques required to maintain trustworthiness.

## Results

Forty families were invited to participate in the research, of which eight parents refused to participate due to various reasons, including lack of satisfaction (three parents), lack of free time

**Table 2. Techniques considered in order to preserve trustworthiness.**

| Credibility | **Investigator triangulation:** The researcher was involved in the field of research long-term before starting and during the project. Immediately after conducting each interview, specific and obvious points of the interview were written in the relevant notes. **Participants triangulation:** The family caregivers were interviewed to triangulate the data source. The participants were invited from different medical centers. **Data collection triangulation:** Semi-structured interviews were conducted, and researcher field notes were kept. Data triangulation involved a paper trail of investigator memos and participant transcripts. **Participant validation:** The participants were asked to confirm the data obtained at the stages of data collection. All participants were allowed to review the audio and/or video records to confirm their experience. Some participants were asked to re-interview by phone if they could make additional comments. The second, third, and fifth authors evaluated and peer-reviewed all the interview transcripts, codes, and categories to guarantee the trustworthiness of the results. |
|---|---|
| Transferability | Direct quotations and a precise and detailed report were provided to promote transferability so the reader could follow the research process. Direct quotes and transferability represented rich explanations and purposeful sampling [12, 20, 21]. The transferability of the study was enhanced by selecting families from diverse educational levels, faiths, economic positions, and socioeconomic classes. |
| Dependability | An external observer checked the research protocol and documented all research steps to ensure the results' dependability. The study's findings were given to six study participants. Also, several participants (adolescents, parents, and therapists) were asked to verify and resolve the interview-generated codes (member check). Participants (adolescents, parents, and therapists) were asked to verify and resolve the interview-generated codes (member check) to ensure the results' validity. The second, third, and fifth authors evaluated and peer-reviewed all the interview transcripts, codes, and categories to guarantee the trustworthiness of the results. |
| Confirmability | All stages of the inquiry were documented to ensure the reliability of the results. Corrective opinions of research members and a re-review of the codes were considered. Also, the research steps and the resulting codes were recorded in detail so others could follow up. Therefore, the method, the coding process, and the list of codes based on the execution date are recorded in separate files. Regarding asymmetry, the theme was determined based on a consensus among the study team members. All codes and categories were reviewed and approved by the authors. The research team regularly met to compare codes, subcategories, and categories. Data saturation was defined as completing all the categories identified by the gathered data with no further data or conceptual material extracted, which determined the sample size. |

(two parents), and unwillingness to talk (three parents). Seventy-four participants were interviewed, including 32 adolescents treated with braces and their parents (27 mothers, five fathers), six orthotists, two physiotherapists, and two spine surgeons. Table 3 includes the participant's characteristics.

Four main categories, 14 categories, and 69 subcategories of 2403 related codes were identified (Fig 1). The main categories' sequences contain the compliance process from the first face with the problem. Time played an effective role in creating milestones and changing adolescents' conditions and compliance rates.

## Accepting the diagnosis

**There is a problem.**　Containing five subcategories (facing the symptoms, discovered by a therapist, being aware, visible spinal deformity, and mother's attention) that emerged from finding a spinal problem. In response to the researcher's question, *"How did you first find out that your spine had a problem?".* The responses referred to some general ways of finding the problem. Some participants stated that having body asymmetry or difficulty sitting, like back pain, made them curious about a problem. Some said their mother had noticed an abnormal spine appearance. In a second way, therapists noticed a deviation in the spine during screening and examination and informed the adolescent parents.

> *A(Adolescent)16. As a teenager, I had trouble sitting comfortably for long periods. So, my mother found Dr. "G" online, and we went to see him for an examination.*

> *A24. I noticed that I had trouble maintaining good posture and had to be reminded by someone to sit up straight. I had also seen children wearing braces at school for similar issues and realized that my angle was more than normal. So, I knew that it was time for me to take action and correct my posture.*

**Seeking for a cause.**　This category consisted of five subcategories (lack of knowledge, parent's searching, attending several physicians, physician's disagreements, and referring to an orthopedist). Initially, as the parents had the least information about the disease, they asked the physician about the cause of the deformity. A physician had referred some families to the

**Table 3. The details of participants.**

|  | Brace type | Girls (Boys) | |
|---|---|---|---|
| **AIS** | Milwaukee brace | 16 (2) | |
|  | TLSO[a] | 5 (1) | |
| **SK** | Milwaukee brace | 4 (3) | |
|  | TLSO | 1 (0) | |
| **Mean Cobb** | AIS | | 37.20˚ ± 11.46˚ |
|  | SK | | 71.8˚ ± 5.9˚ |
| **Mean age** | 14.2 ±3.5 years | | |
| **Mean brace-wearing time** | 8.5±1.4 months | | |
| **Mothers** | 27 | | |
| **Fathers** | 5 | | |
| **Orthotists** | 6 | | |
| **Physical Therapists** | 2 | | |
| **Spine Surgeons** | 2 | | |

[a]Thoracolumbosacral orthoses

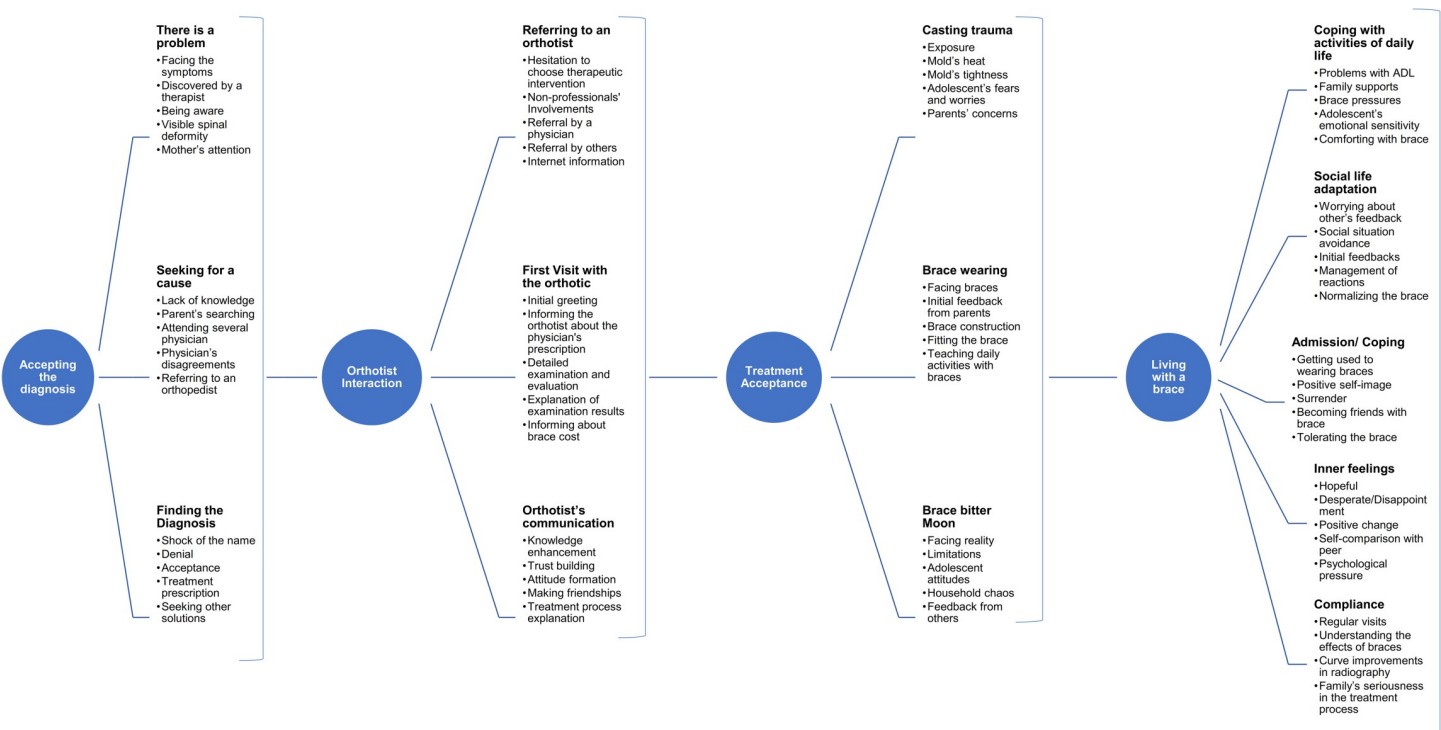

**Fig 1. Categories were extracted from the interview analysis during the brace compliance process.**

orthopedic surgeon; however, other families were confused and hesitant because of attending to several physicians. Finally, the patient's disease was diagnosed accurately after referring the patient to the orthopedic surgeon.

*M(Mother)15. I was completely unaware of the situation since it was an unprecedented case, and as far as I knew, there was no remedy available for it.*

*A5. Dr. "R" initially advised me that there was no issue and that I should continue with my life without any follow-up. However, when I saw Dr. "G," I was expecting him to say the same thing, but he mentioned that my spine had a curve of 28 degrees and that I needed to wear a brace for 23 hours a day.*

**Finding the diagnosis.** Consisted of five subcategories (shock of the name, denial, acceptance, treatment prescription, and seeking other solutions). Due to the lack of any prior information about the disease, families get shocked after hearing the name of the disorder, and many questions and concerns form in their minds. The adolescents reported feeling guilty, anxious, upset, and scared after the orthopedist's speech. The first reaction after hearing the orthopedist's diagnosis was denial. The mothers said that usually, the fathers tried to deny the disease and insisted that there was no issue and that the child would get better by growing. Some parents referred to other orthopedists to ensure. The orthopedic surgeon briefly explained the treatment process, duration, and efficacy. In this stage, the bracing concept was unknown and unimaginable to adolescents, who had many questions. Before accepting brace treatment, about 30% of families sought other solutions. It included exercise therapy alone, corrective movements, and thinking about surgery after skeletal puberty.

*A8. I suddenly panicked when I noticed something new happening in my body. It was a scary experience for me as it was the first time I had ever felt this way.*

*M6. I shared the news with my spouse, but we were both initially in disbelief. We had no idea that such a thing even existed.*

### Orthotist interaction

**Referring to an orthotist.** Referring to an orthotist is often done by a physician and others like exercise trainers, acquaintances, and the internet. Five subcategories (hesitation to choose therapeutic intervention, non-professionals' involvements, referral by a physician, referral by others, internet information) developed the Referring to an orthotist category.

*A25. My father didn't like the idea of the brace. He told me to sit up straight without it. However, I struggled to maintain proper posture and felt frustrated. Eventually, my mom bought me a brace, which helped me to sit up normally.*

*A16. I was advised to look up information online, and I discovered the Orthotic Center through my search.*

**First visit with the orthotic.** The initial relationship with the family was formed during the first visit with the orthotic. This category was developed into five subcategories (initial greeting, informing the orthotist about the physician's prescription, detailed examination and evaluation, explanation of examination results, and informing about brace cost).

According to the orthotists' statements, in the first meeting, families come to the orthotists with shock, confusion, sadness, and anxiety. Orthotists emphasized that after creating a sense of safety, examining and evaluating adolescents is necessary. Orthotists said families usually have many questions and ambiguities about the nature of the brace, its material, weight, cost, how long it takes to wear the brace, and how effective it would be. In this stage, the orthotists provide accurate information, answer parents' worries, explain treatment duration, brace function, and the necessity of exercise with brace wearing, and improve families' information regarding spinal deformities and brace treatment. The parents also stated that in the first meeting with the orthotists, they had reduced their stress and anxiety by creating intimacy, offering solutions, and empathy with the family.

*O(Orthotist)3. During our initial meeting, we reviewed the radiographic images of the spine and discussed the device we would be using in our case.*

*O4. I assess the patient's condition and conduct a physical exam. I receive a report and a photo as part of the process. The first time I viewed the photo, I also reviewed the prescription.*

**Orthotists' communication.** This category emerged from these subcategories (knowledge enhancement, trust building, attitude formation, making friendships, and treatment process explanation). After good communication was established with the orthotist and the families received the answers accurately, trust in the orthotist was formed. During this stage, the parents' minds formed an accurate attitude toward the purpose of brace treatment, brace efficacy, the probability of preventing surgery, and the necessary conditions for the effectiveness of brace treatment.

In addition to answering families' questions, orthotists explain the prevalence of the disease and the effectiveness of timely treatment to reduce family worries and anxieties. Especially for major curves, if the controlling effect of the brace on the curve were not well explained to the family, their expectation of brace performance during treatment would be unrealistic.

*O6. During the initial evaluation and fitting, I provided a thorough explanation of the patient's condition, the use of the brace during physical activities, and the benefits that the brace would offer.*

*M32. The doctor was very respectful and patient during our appointment. He took the time to answer all of our questions and concerns regarding the brace and how it would affect "S." He provided clear and realistic explanations and instructions, which we found helpful and reassuring.*

## Treatment acceptance

**Casting trauma.** This category has created a relatively unpleasant experience for adolescents and contains these subcategories (exposure, mold's heat, mold's tightness, adolescent's fears and worries, and parents' concerns). They felt shame and embarrassment as they had to be exposed during casting. The heat generated by the plaster and the high rigidity of the cast until it formed on the body are also mentioned as the causes of this annoying experience.

*A9. I had an unpleasant experience. The presence of mold made me feel uncomfortable. I didn't like it at all, especially because I ended up with something sticky on my body. Additionally, when Dr. "K" instructed me to stare at one spot and not move, it wasn't a pleasant experience either.*

*O5. We conducted body scans of the adolescents who had to expose their trunks entirely. However, they often felt embarrassed during the process.*

**Brace wearing.** This category emerged from these subcategories (facing braces, initial feedback from parents, brace construction, fitting the brace, and teaching daily activities with braces).

The first time the adolescents encountered a half-ready brace for adjustment, they often felt confused, asking, "*How can I stand this*?". One of the adolescents said I felt excitement, and one of them said, "*I feel like wearing a horse saddle*," after wearing a Milwaukee basket. During the brace fit and adjustment, the orthotist answered adolescents' questions about daily activities with brace wearing. After the final brace delivery, the adolescents were taught how to sleep, get in the car, the right position to study, what activities to limit, how to face others, and how often to clean the brace. The family was also urged to return for resuscitation if their children felt excessive pressure on the pads and were severely bothered by the brace.

*A17. Getting into the car was quite challenging for me initially. I felt quite nervous and overwhelmed.*

*A26. I remember feeling really nervous about wearing a brace. I had seen other kids wearing them at school and knew how difficult it could be. At the time, I didn't think I could handle it.*

**Brace bitter moon.** Brace bitter Moon emerged from these subcategories (facing reality, limitations, adolescent attitudes, household chaos, and feedback from others). After returning home, the adolescent gradually faced changes in circumstances. One of the main complaints of the teenagers was that they could not sleep during the first nights with the brace, and they mainly cited the feeling of numbness, suffocation, and shortness of breath due to the pressure of the brace and the change of sleeping position to supine. It took about a week to get used to these conditions while sleeping. Difficulty eating was the next problem mentioned in the first days of wearing the brace.

During the first few days, daily activities became hard for the adolescents, and they had to devote more time to each activity. In some cases, they could not work independently and needed family support. The adolescents realized that their simple daily activities, such as eating, going to the bathroom, studying, sitting, and getting up from the sofa and chair, would not be easy anymore. The adolescents became upset, and some parents said their child was nervous at this period.

Adolescents and their parents stated that the first month of brace treatment was the most difficult and unpleasant period. During this month, the families experienced psychological shock because of wearing the brace. During this period, half of the mothers suffered from anxiety, the effects of which were transmitted to the adolescent.

*A7. I found myself lying on my back like a turtle, and no matter how hard I tried, I couldn't get back on my feet.*

*M1. The initial days were extremely tough for both "G" and me. It felt as if a complete stranger had entered our home, leaving us in a state of despair and helplessness. The feeling was terrible and overwhelming.*

## Living with a brace

During the first few weeks of wearing the brace, the children said they experienced problems that made them feel chaotic and upset. Their complaints included dependency on the parents or siblings for personal tasks, low-quality night sleep at the beginning, difficulty in recreational activities (like dancing, playing instruments, and playing with peers), slowing down for personal activities, and thus spending more time doing daily simple tasks.

**Coping with activities of daily life.** The first category of living with a brace was coping with activities of daily life (ADL) that included five subcategories (problems with ADL, family supports, brace pressures, adolescent's emotional sensitivity, and comforting with brace). Regarding the lifestyle changes following brace wearing and devoting two hours a day to exercise, time management was one of the most important challenges adolescents faced during brace treatment. For this, they could spend less time studying and doing homework.

*A22. Over time, I became accustomed to wearing the brace. I slept more soundly and felt more at ease.*

*A7. I initially felt that it was too tight. Standing up straight and walking was quite uncomfortable. However, it gradually improved over time.*

**Social life adaptation.** This category contained five subcategories (worrying about others' feedback, social situation avoidance, initial feedback, management of reactions, and normalizing the brace).

According to the adolescents and their parents' statements, early in the brace treatment period, the adolescents were very worried about being seen and judged by others. It was clear from the parents' words that the more this concern was for the parents, the more important it was for the children. Most adolescents stated that they felt different from others during the treatment period, even when they accepted the brace mentally. This puts a lot of psychological burden on the children, and it was a question in their minds, *"Why did this happen to me?"*. Early in the treatment period, adolescents preferred not to be present in family gatherings with the brace.

In response to the question, *"Why did you avoid attending publics with a brace?"*, children mentioned that they were reluctant to attract others' attention, and in that case, they would be embarrassed. Also, some said they do not want to discuss wearing braces, even with close friends. The adolescents reported that over time, after brace acceptance and learning what to say in response to others' questions and curiosities, their anxiety about attending gatherings and being seen with the brace decreased. The questions people usually ask when seeing an adolescent with a brace include two main parts: 1. What is this that you wear? 2. Why should you wear this device?

Some adolescents referred to the same reasons that their orthotists or parents said. Almost all the adolescents said that they give as concise answers as possible to people and have no interest in explaining the subject. In response to the time of wearing the brace, they did not say a specific time, or even one of the adolescents replied, *"I have to wear it for a month, and then I would get better."* In half of the cases, adolescents said their peers gave them titles such as "hanger, iron, scaffolding, skeleton, armor," but most teens said they often did not bother us. After learning about the purpose of wearing a brace, most adolescents noted that their peers became accustomed to it and did not look at it as a strange object, which no longer attracted their attention.

*A7. Kids always seemed to find my brace fascinating, but I never felt comfortable explaining it all the time. While I was happy to answer their questions, I just didn't enjoy discussing it too much.*

*A32. It depends on the context. If someone was making fun of me for wearing a brace that I need for my health, I would feel upset and misunderstood. However, if someone was asking out of curiosity and genuinely interested in why I need the brace, then I would be happy to explain and grateful for their thoughtful question.*

**Admission/coping.**   Another category, emerged of five subcategories (getting used to wearing braces, positive self-image, surrender, becoming friends with the brace, and tolerating the brace)

The adolescents said they mostly found various ways to adjust to the new situation after a month and got used to wearing the brace, even for sleeping. They used adjustable tables to do their homework and had no problem eating due to their habit of posing with braces.

They also said that over time, they were able to have good time management for daily activities, including doing exercises, taking a quick shower, and doing favorite activities. Most said wearing the brace is much easier than it looks. Most adolescents gradually came to terms with the new conditions created by wearing a brace for one to three months. The adolescents got used to their braces during the period; they resembled the brace as a friend and had become friends with them. Most adolescents also stated that *"the brace became a part of me after a while."* The adolescents became more comfortable wearing the brace after adapting to daily life

and social interactions. *"I don't feel like I'm wearing anything extra right now,"* said one teenager.

> *A14. Now, I rarely think about my brace problems. When I do, it becomes increasingly difficult to manage.*

> *A32. When I first got my brace, it felt separate from me, but now I see it as a part of my body.*

**Inner feelings.** Inner feelings contained five subcategories (hopeful, desperate/disappointment, positive change, self-comparison with peers, psychological pressure). The adolescent's inner feelings were not constant during the brace treatment but fluctuated between positive and negative feelings. After going through a period of shock and accepting the brace, the adolescents hoped to be treated by the brace. They also experienced frustration in some periods, leading to confusion and despair. One of the important events in the treatment period was the results of spinal radiographs showing treatment improvement. Adolescents who have reached the stage of full acceptance of the treatment plan and have coped well with it have indicated during the interview (positive feedback) that they have received positive feedback from family and others after starting treatment.

> *A16. I have faith in this brace. It has the potential to work within four to six months, give or take a little time. Because of this, I am motivated to use the brace more often.*

> *A9. After wearing the brace, I can breathe well and sing, my voice has improved greatly, and I don't have as much problem with my violin. I can play my violin better than I used to play it before.*

**Compliance.** Eventually, in those families that had achieved brace adherence, compliance contained four subcategories: (regular visits, understanding the effects of braces, curve improvements in radiography, and family's seriousness in the treatment process)

Most families receiving treatment followed the regular brace adjustment visits. Therefore, the pressure of the pads was adjusted in time, and as a result, the place of the brace was not annoying for the children, and the brace covers were replaced in time. The parents who were committed to the regular follow-ups said that during periodic follow-ups, the orthopedist and orthotist objectively explained the therapeutic effects of wearing braces and documented them on the radiograph. Therapists also provided realistic information about how much each curve changed during the treatment period and how effective the treatment plan was in controlling the curve progression. One physiotherapist acknowledged that the first follow-up and radiograph were very important for the patient to see the results of wearing the brace and alleviate some of the family's concerns.

After these steps, adolescents were assured of treatment acceptance over time and coping with individual and social issues. Brace was no longer an annoying mental concern for them, and they focused on brace treatment with a focus on recovery and prevention of surgery.

> *A25I went to the center every two weeks since I got the brace. They would adjust my brace, and I would do exercises there.*

> *A32. I am fully convinced and justified about the purpose of wearing the brace. The words of others have helped me set a goal for myself.*

## Discussion

This study aimed to explore the brace compliance process in adolescents with spinal deformities using a qualitative approach. Hence, interviews were conducted to consider the importance of brace acceptance in treating spinal deviations and to understand the direct experiences and views of the people involved. Following the interview analysis, the brace compliance formation process appeared. Because wearing a brace is stressful and restricts daily movements [22], patients usually stop wearing the brace and discontinue the treatment [8, 9].

Based on the analysis of the interviews in this study, the first step in the brace compliance process is for adolescents to accept that there is a problem with their spines. Like other chronic diseases, patients with spinal deformities experience shock after learning of the physician's diagnosis.

The present study's findings showed that the moment of finding out about the disease was the first critical time that the adolescents and their families faced. Most families reported that they were shocked and worried after the physician announced the spinal deformity, which was mostly caused by the lack of information about the scoliosis. Previous studies have shown that the correct formation of the brace compliance process in a patient affects the treatment result [10, 23].

According to previous studies, diseases that change the structure of the trunk can also change the body image, and reactions to such changes are influenced by various factors, including family attitudes and cultural issues [24].Reichel et al. [25] noted in their study that following the diagnosis of spinal deformities in adolescents and the start of brace therapy, a stressful situation is created for adolescents; this happens in a situation where uncertainty about the success of treatment and changes in lifestyle also put pressure on the patient.

The results indicated that in the first visit to the brace clinic if the orthotist's description is incomplete and the adolescent is not informed of wearing the brace's purpose and function, the compliance process will be stopped at the phase of getting the brace. In the first session, it is necessary to reduce the tension and psychological pressure on the adolescents to increase their hope and motivation to start treatment with braces.

Interviews showed that how the orthotist communicates with the adolescent is crucial, and successful communication requires spending enough time to get to know the patients, empathize, understand each other, and answer their questions. Cognitive empathy produces healthier results, and research authors recommend interventions that reduce emotional empathy and increase cognitive empathy in therapists [26].

As a result, it seems that the sense of trust and forming an effective relationship between the adolescent and the therapist, especially the orthotist, requires time and energy. It was reported that families should be informed precisely about the importance of wearing a brace, as expected by the orthopedic surgeon [27]. Another role that orthotists should play is correcting the previous information and background of the family and the adolescent regarding spinal deviations and treatment with braces.

Going through this phase, the adolescents complained about the situation despite the desire to improve the deformity, and some of them said they cried. All the adolescents said that the most challenging time with the brace was the first and second months, after which they coped.

Since molding is one of the first stages of brace treatment, adolescents experience the greatest amount of worry and mental pressure. The orthotists stated that, at the beginning of the treatment period, when the adolescent had to undress in the presence of the therapist, it was a painful feeling for them. These findings were also confirmed in the study of Grantham et al. that the annoying feeling of embarrassment due to being naked during the diagnosis shows

the extreme sensitivity of the adolescent towards the changing body of puberty and the need to respect privacy, and it makes maintaining respect more prominent [28].

After the family and adolescent come to terms with scoliosis, this phase begins with the new shock of wearing a brace. Almost all the adolescents stated that the most difficult time with braces was the first one to two months, and then they coped with the situation.

The key concern of adolescents aged 12 to 18 years old is to resolve the identity crisis and confusion and enter the youth-adulthood period with a clear understanding of who and what they are [29]. At the same time as struggling with this psychosocial disturbance, the adolescent's body undergoes significant physical changes. On the other hand, during adolescence, there is a conflict between a person's lack of need for his parents—because this need challenges their idea of himself as an independent person—however, when using a brace, the adolescent strongly needs parental support [30].

Some problems that adolescents and their parents mentioned at the beginning of the treatment period were poor sleep and difficulty sleeping at night with a brace, shortness of breath, stomach pain after eating, and difficulty in sitting and standing up. Another notable issue was that at the beginning of the treatment period, when the adolescent has not yet come to peace with his new situation, he is mentally more vulnerable to any unpleasant feedback, even extreme sympathy from close relatives such as his grandmother and grandfather.

The limitations of running and playing with peers during brace treatment were repeatedly mentioned by the younger age group (10–13 years). The major challenge faced by adolescents in the age group of 13–16 years, and to the best of our knowledge not mentioned in previous studies, was the problems created following brace treatment and exercise for adolescent studying issues.

Several mothers stated that the number of disputes and arguments they had with their teenage children increased, and it became more challenging to communicate with them during the brace treatment period. According to previous studies, raising a child with scoliosis in the stages of treatment and management of scoliosis can be a complex process [31] and stressful [32], and considering spinal deviations, it is not an uncommon experience. Another part of the problems the adolescents complained about in the interviews is related to the physical conditions created for them by wearing braces at school.

Among the things that almost all adolescents agreed on as the difficulties of wearing a brace were the heat inside the brace, restrictions on movements, especially bending, and restrictions on choosing clothes. Over time and after coping with these issues, the pressures applied by the brace, which sometimes lead to the formation of wounds, were mentioned among the difficulties of wearing the brace.

The embarrassment of being present among people with braces is greater in adolescents between 13–16 years old. Sapountzi et al. [33] noted in their study that adolescents do not want to look different from their peers; although wearing a brace is not painful, wearing it for patients is annoying. Although the most important social relations with peers are outside the home during adolescence [34, 35], most adolescents in the present study avoided attending gatherings and parties as much as possible at the beginning of the treatment period. They preferred that few people see them in new conditions. It was also reported in previous studies that adolescents use some solutions for their concerns about their appearance and body image and hide physical abnormalities, such as wearing loose clothes, avoiding social interactions and special situations, and trying to avoid inducing the feeling of a different appearance [33].

According to Asher et al.'s study, this period leads to decreased physical functions, changes in appearance, long-term dependence on medical professionals, and changes in the adolescent's life perspective [36]. The analysis of interviews with adolescents and their parents showed that during the period of treatment with braces, adolescents experience severe

emotional fluctuations. Adolescents stated about experiencing feelings of guilt, embarrassment, frustration, sadness, and loneliness in the process of brace compliance. Therefore, the emotional support of those around them is necessary for adolescents using braces [37]. Sapountzi et al. believed that the bio-psychological and social effects of scoliosis and brace therapy in adolescents, due to the psychological difficulty of this age group, can cause stress in them [33]. Also, the study of Asada et al. [38] reported that "feeling anxious about how to be perceived by others" caused stress caused by brace treatment in patients with scoliosis. It seems that the unpleasant experience of braces is mainly caused by psychological burden rather than the pressure of corrective pads [39].

Brace compliance is an endurance event that lasts several months and sometimes years. Clinical support should go beyond prescribing a brace and monitoring compliance, including emotional and psychological support. In the present study, according to the statements, the adolescent gets tired after a few months due to the long treatment period and needs re-motivation. Some adolescents stated that their bodies got used to wearing braces, especially for sleeping. In cases where the adolescent's sibling had also experienced brace treatment, the adolescent accepted the new conditions sooner. Adolescents must receive logical and scientific arguments to understand the importance of wearing braces. Therefore, determining the intervals of regular visits in a notebook seems useful for encouraging and continuous communication between the orthotist and the adolescent and adjusting the pressure of the brace pads. Since brace acceptance is poor, previous studies recommend that full-time wear (24 hours a day, 7 days a week) be "prescribed" even with the inevitable periods of non-acceptance [40, 41]. On the other hand, the analysis of the interviews showed that the extreme strictness of the parents and successive reminders to wear braces regularly have the opposite results and make the adolescent more stubborn. In our study, the therapists emphasized that to maintain the adolescent's morale, the feeling of being sick should not be instilled in the adolescent, and he should not be treated like a person who has a special disease; on the contrary, it is better to focus on the effect of wearing a brace on the appearance, such as becoming healthier.

## Limitation

It should be declared that during this study, the authors did not have any intervention or authority to prescribe and suggest the type of brace treatment, and only participants from the three orthotic and prosthetic centers were invited. According to recent studies and the effectiveness of asymmetric design braces [42, 43], it is suggested that in the future, a study with a qualitative approach should be conducted on the experience of wearing asymmetric design braces of adolescents with spinal deformities.

## Conclusion

The current qualitative research found that the long treatment period with a brace creates extremely tiring conditions for adolescents, which will ultimately cause non-compliance with the brace. Accepting brace therapy is a big challenge, given the significant change it causes in an adolescent's appearance. Adolescents are involved in disease and treatment for a long time without any preparation, which affects their whole individual and social lives. The psychological burden after the diagnosis and the start of treatment may be higher than the usual capacity of the adolescent at this age. Also, staying away from a series of recreations and physical activities that adolescents like makes excuses and unwillingness to wear braces regularly.

The adolescent's transition from the difficult initial conditions of brace therapy and the formation of the compliance process requires deep understanding, special attention, and support in all aspects from family, treatment team, and friends.

## Supporting information

**S1 Table. Consolidated criteria for reporting qualitative studies (COREQ).** The 32-item checklist of Consolidated criteria for reporting qualitative studies (COREQ) completed for this study.
(DOC)

## Acknowledgments

The authors would like to thank the participants in the interviews for their time and interest in this project.

## Author Contributions

**Conceptualization:** Mohammad Kamali, Hadi Ranjbar, Mojtaba Kamyab.

**Data curation:** Faezeh Ghorbani, Hiva Razavi, Taher Babaee.

**Formal analysis:** Faezeh Ghorbani, Hadi Ranjbar, Hiva Razavi.

**Funding acquisition:** Faezeh Ghorbani.

**Investigation:** Faezeh Ghorbani, Hadi Ranjbar, Mojtaba Kamyab, Taher Babaee.

**Methodology:** Faezeh Ghorbani, Mohammad Kamali, Hadi Ranjbar.

**Project administration:** Faezeh Ghorbani, Mohammad Kamali, Hadi Ranjbar.

**Resources:** Faezeh Ghorbani.

**Supervision:** Mohammad Kamali, Hadi Ranjbar.

**Validation:** Faezeh Ghorbani, Hadi Ranjbar, Taher Babaee.

**Visualization:** Faezeh Ghorbani, Mohammad Kamali, Hadi Ranjbar, Mojtaba Kamyab, Taher Babaee.

**Writing – original draft:** Faezeh Ghorbani, Hiva Razavi, Taher Babaee.

**Writing – review & editing:** Faezeh Ghorbani, Mohammad Kamali, Hadi Ranjbar, Mojtaba Kamyab, Hiva Razavi, Taher Babaee.

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
