## [Decision Letter · Decision Letter 0]

25 Mar 2024

PONE-D-24-08333Brace compliance process in adolescents with spinal deformities: a qualitative studyPLOS ONE

Dear Dr. Ranjbar,

Thank you for submitting your manuscript to PLOS ONE. After careful consideration, we feel that it has merit but does not fully meet PLOS ONE’s publication criteria as it currently stands. Therefore, we invite you to submit a revised version of the manuscript that addresses the points raised during the review process.

We look forward to receiving your revised manuscript.

Kind regards,

Raffaele Vitiello

Academic Editor

PLOS ONE

Journal Requirements:

4. We note you have included a table to which you do not refer in the text of your manuscript. Please ensure that you refer to Table 2 in your text; if accepted, production will need this reference to link the reader to the Table.

Additional Editor Comments:

** **According to reviewers a major revision is needed before the publication.

Reviewers' comments:

Reviewer's Responses to Questions

**Comments to the Author**

1. Is the manuscript technically sound, and do the data support the conclusions?

Reviewer #1: No

Reviewer #2: Partly

2. Has the statistical analysis been performed appropriately and rigorously? 

Reviewer #1: N/A

Reviewer #2: N/A

3. Have the authors made all data underlying the findings in their manuscript fully available?

Reviewer #1: No

Reviewer #2: Yes

4. Is the manuscript presented in an intelligible fashion and written in standard English?

Reviewer #1: Yes

Reviewer #2: Yes

5. Review Comments to the Author

Reviewer #1: Dear editor,

Thank you for your kind invitation to review the study entitled "Brace compliance process in adolescents with spinal deformities: a qualitative study".

I congratulate the authors for the effort they put into this study. I think it is a very important study, but there is a significant methodological problem.

Asymmetric braces are effective in the treatment of scoliosis and Milwaukee brace has not been used in the treatment of scoliosis for about 20 - 25 years. In this sense, I strongly recommend the authors to update their information.

“The Milwaukee brace has a negative influence on quality of life, physiological problems, sleep disorders and many disadvantages in the areas of body image. However, the most significant of these is its appearance, which reduces its acceptance due to cosmetic deterioration and discomfort “

It is known that braces that are designed asymmetrically, with spaces where the body can go opposite the corrective zones, are effective.

Ref: Weiss HR, Çolak TK, Lay M, Borysov M. Brace treatment for patients with scoliosis: State of the art. S Afr J Physiother. 2021 Oct 26;77(2):1573. doi: 10.4102/sajp.v77i2.1573. PMID: 34859162; PMCID: PMC8603182.

Weiss, H. R., & Kleban, A. (2015). Development of CAD/CAM Based Brace Models for the Treatment of Patients with Scoliosis-Classification Based Approach versus Finite Element Modelling. Asian spine journal, 9(5), 661–667. https://doi.org/10.4184/asj.2015.9.5.661

Weiss, H. R., Lay, M., Seibel, S., & Kleban, A. (2021). Ist eine Verbesserung der Behandlungssicherheit in der Korsettversorgung von Skoliosepatienten durch Anwendung standardisierter CAD-Algorithmen möglich? [Is it possible to improve treatment safety in the brace treatment of scoliosis patients by using standardized CAD algorithms?]. Der Orthopade, 50(6), 435–445. https://doi.org/10.1007/s00132-020-04000-9

It is also unclear why the researchers used a Milwaukee brace and not a kyphosis brace in patients with kyphosis.

I suggest the authors to redesign their study. In this age group, it is important that they receive appropriate treatment, especially during the period of rapid progression of scoliosis.

Although the efficacy levels of scoliosis braces are reported differently in different studies, the most effective braces are asymmetric braces.

Furthermore, the authors wrote that cases between 10 and 14 years of age would be included, but the average age presented in the table is older than 14 years. When the standard deviation is added, it is seen that even 17-year-old individuals are included.

The authors did not present the Cobb angles of the patients.

It was reported that children who had just started using a brace would be included in the study, but it was reported that children had been using a brace for more than 8 months.

I suggest the authors not to use abbreviations in the abstract.

Reviewer #2: The authors present an interesting on a less debated argument. The qualitative approach in considering brace compliance in AIS and SK is very unusual and give interesting and useful information.

In the present form the manuscript is not suitalble for publication and need some clarifications and revisions.

The paper is too long and of difficult reading. The methods and results section are unproportionally long if compared to discussion and introduction

The qualitative approach is main purpose of the study but it could be useful to summarize questions of the review perhaps a table reporting all the question could de add to clarify.

In reporting the obatained results a more organic approach could be useful perhaps categorising the answers and giving some final picture of the collected results.

Have the authors noted a difference in compliance between the TLSO and Milwkee users? If yes have they found a clear reason?

The discussion section has to unique and no divided in section

6. PLOS authors have the option to publish the peer review history of their article (what does this mean?). If published, this will include your full peer review and any attached files.

Reviewer #1: No

Reviewer #2: No

---

## [Author Response · Author response to Decision Letter 0]

18 May 2024

Journal Requirements:

Response. Many thanks; we rechecked the PLOS ONE style templates and revised some points.

Response. Thanks for your advice. Hiva Razavi, one of the authors, has thoroughly edited the manuscript. Both *supporting information* and *manuscript* files will be submitted. 

Response. As you requested, the corresponding author’s ORCID information was updated on the author's page. 

4. We note you have included a table to which you do not refer in the text of your manuscript. Please ensure that you refer to Table 2 in your text; if accepted, production will need this reference to link the reader to the Table.

Response. Many thanks for your attention. Table ‘numbers and captions have been revised and checked. 

Response. The captions of the Supporting Information files were added at the end of the manuscript.

 

Reviewer# 1: Dear reviewer: thank you for your thoughtful and thorough review of our manuscript.

Dear editor,

Thank you for your kind invitation to review the study entitled "Brace compliance process in adolescents with spinal deformities: a qualitative study".

I congratulate the authors for the effort they put into this study. I think it is a very important study, but there is a significant methodological problem.

Asymmetric braces are effective in the treatment of scoliosis and Milwaukee brace has not been used in the treatment of scoliosis for about 20 - 25 years. In this sense, I strongly recommend the authors to update their information.

“The Milwaukee brace has a negative influence on quality of life, physiological problems, sleep disorders and many disadvantages in the areas of body image. However, the most significant of these is its appearance, which reduces its acceptance due to cosmetic deterioration and discomfort “

It is known that braces that are designed asymmetrically, with spaces where the body can go opposite the corrective zones, are effective.

Ref: Weiss HR, Çolak TK, Lay M, Borysov M. Brace treatment for patients with scoliosis: State of the art. S Afr J Physiother. 2021 Oct 26;77(2):1573. doi: 10.4102/sajp.v77i2.1573. PMID: 34859162; PMCID: PMC8603182.

Weiss, H. R., & Kleban, A. (2015). Development of CAD/CAM Based Brace Models for the Treatment of Patients with Scoliosis-Classification Based Approach versus Finite Element Modelling. Asian spine journal, 9(5), 661–667. https://doi.org/10.4184/asj.2015.9.5.661

Weiss, H. R., Lay, M., Seibel, S., & Kleban, A. (2021). Ist eine Verbesserung der Behandlungssicherheit in der Korsettversorgung von Skoliosepatienten durch Anwendung standardisierter CAD-Algorithmen möglich? [Is it possible to improve treatment safety in the brace treatment of scoliosis patients by using standardized CAD algorithms?]. Der Orthopade, 50(6), 435–445. https://doi.org/10.1007/s00132-020-04000-9

- It is also unclear why the researchers used a Milwaukee brace and not a kyphosis brace in patients with kyphosis. 

Response. Many thanks for highlighting this point. We agree that using modern lower-profile orthoses has better psychosocial outcomes for individuals with AIS or kyphosis. However, in this qualitative study, our aim was not to evaluate the superiority of one orthosis to another. Moreover, the Milwaukee brace is the most prescribed orthosis in our country for no-surgical treatment of adolescents with spine deformities. Although there are some orthotists in our country who can manufacture asymmetric braces, they recently received their certification. Therefore, the number of prescriptions for the Milwaukee brace is still high. To address your valuable comment, we have added some sentences in the limitation section of the manuscript with the supporting references you provided. Regarding kyphosis patients, the Milwaukee brace was prescribed because all the included cases had a large kyphosis curve with an apex of above T8. 

- I suggest the authors to redesign their study. In this age group, it is important that they receive appropriate treatment, especially during the period of rapid progression of scoliosis. Although the efficacy levels of scoliosis braces are reported differently in different studies, the most effective braces are asymmetric braces.

Response. Many thanks for your attention. You are absolutely right, and we also acknowledge the effectiveness of asymmetric braces based on previous studies. However, the purpose of this study was not to investigate the superiority of one brace to another. In this study, our aim was to investigate the process of brace compliance based on the experiences of adolescents who are using a brace. In this study, the authors did not have any intervention or authority to prescribe and suggest the type of brace, and they only invited interviewees from the three orthotic and prosthetic centers to be included. Since the treating spine surgeons refer to their textbook to prescribe the best orthotic option for adolescents with spine deformities, conventional brace, such as the Milwaukee brace and Boston brace, are our country's most prevalent prescribed orthosis. It should be noted that some orthotists can manufacture the asymmetric braces in our country, and future studies can be designed to evaluate the superiority of asymmetric braces to symmetric braces. Moreover, this will be a fruit area for researchers to assess the lived experiences of adolescents using an asymmetric brace and compare their experiences with those using a symmetric brace. Based on your suggestion, we mentioned this issue as a limitation in the discussion section and suggested conducting studies on adolescents using modern and up-to-date braces. Please see page 24, lines 491-494. This is clarified in the limitation section of the manuscript. 

- Furthermore, the authors wrote that cases between 10 and 14 years of age would be included, but the average age presented in the table is older than 14 years. When the standard deviation is added, it is seen that even 17-year-old individuals are included.

Response. Thanks for your comment. Actually, all participants were 10-14 years old at the time of deformity diagnosis and starting the brace treatment. We invited those with experience using a brace for at least three months for the interviews. As a result, the average age of the participants during the interviews was more than 14 years. The participants told the interviewer about their past and experiences from the days of deformity diagnosis and the beginning of the brace treatment to the day of the interview. Based on your comment and to clarify this, some corrections were made in the manuscript on Page 5, line 89.

- The authors did not present the Cobb angles of the patients.

Response. Many thanks for highlighting this point. As per your suggestion, the mean Cobb angle at the time of referring to the orthotic center and beginning braces treatment was added to Table 3.

- It was reported that children who had just started using a brace would be included in the study, but it was reported that children had been using a brace for more than 8 months.

Response. It has been stated that “the study required adolescents who had undergone brace treatment for more than three months.” The participants told the interviewer about their past and experiences from the days of deformity diagnosis and the beginning of the brace treatment to the day of the interview.

- I suggest the authors not to use abbreviations in the abstract.

Response. Many thanks for your comment. The suggested changes have been made in the abstract section, and the abbreviations have been eliminated.

Reviewer #2: Dear reviewer: thank you for your thoughtful and thorough review of our manuscript.

The authors present an interesting on a less debated argument. The qualitative approach in considering brace compliance in AIS and SK is very unusual and give interesting and useful information.

In the present form the manuscript is not suitalble for publication and need some clarifications and revisions.

- The paper is too long and of difficult reading. 

Response. Thanks for your valuable comment. We tried to summarize the text as much as possible. Moreover, some of the quotations have been eliminated.

- The methods and results section are unproportionally long if compared to discussion and introduction

Response. Many thanks for highlighting this point. Based on the journal's guidelines, we have designed the method section and reported the results per the Consolidated criteria for reporting qualitative research (COREQ) checklist. Considering that all checklist items should have been included in the manuscript, the method and result sections are long. However, based on your comment, we have summarized these sections as much as possible. The participants' quotations have been reduced to reduce the content volume of the results.

- The qualitative approach is main purpose of the study but it could be useful to summarize questions of the review perhaps a table reporting all the question could de add to clarify.

Response. Thanks for paying attention to this issue. The guide questions used in the interviews were prepared and submitted as a supplementary file. Based on your opinion, the questions are now in the table 1 and added through the manuscript on pages 3 and 4.

- In reporting the obatained results a more organic approach could be useful perhaps categorising the answers and giving some final picture of the collected results.

Response. Many thanks for highlighting this point. Considering that the design of our study was qualitative, the content analysis of the results led to code extraction, which was further reported as sub-categories and categories formation. The text of the interviews was studied, and their results were coded, which led to the creation of classes and concepts. Therefore, changing how the results are reported may not be in line with created categories and sub-categories. However, the results summary is presented graphically in Figure 1 to address your valuable comment.

- Have the authors noted a difference in compliance between the TLSO and Milwaukee users? If yes have they found a clear reason?

Response. Thanks for your comment. Since the present study was designed as a qualitative study, no statistical analysis was applied. Therefore, we could not accurately report whether there was any difference between these two braces.

- The discussion section has to unique and no divided in section

Response. The suggested changes have been made in the discussion section. In this revised version of the manuscript, the discussion sections have been presented as an integrated part.

---

## [Decision Letter · Decision Letter 1]

3 Jun 2024

Brace compliance process in adolescents with spinal deformities: a qualitative study

PONE-D-24-08333R1

Dear Dr. Ranjbar,

We’re pleased to inform you that your manuscript has been judged scientifically suitable for publication and will be formally accepted for publication once it meets all outstanding technical requirements.

Kind regards,

Raffaele Vitiello

Academic Editor

PLOS ONE

Additional Editor Comments (optional):

Reviewers' comments:

Reviewer's Responses to Questions

**Comments to the Author**

1. If the authors have adequately addressed your comments raised in a previous round of review and you feel that this manuscript is now acceptable for publication, you may indicate that here to bypass the “Comments to the Author” section, enter your conflict of interest statement in the “Confidential to Editor” section, and submit your "Accept" recommendation.

Reviewer #2: All comments have been addressed

2. Is the manuscript technically sound, and do the data support the conclusions?

Reviewer #2: Yes

3. Has the statistical analysis been performed appropriately and rigorously? 

Reviewer #2: N/A

4. Have the authors made all data underlying the findings in their manuscript fully available?

Reviewer #2: Yes

5. Is the manuscript presented in an intelligible fashion and written in standard English?

Reviewer #2: Yes

6. Review Comments to the Author

Reviewer #2: satisfactory revision , all the requested points have been clarified , the general reading of the manuscript is more comprehensible.

7. PLOS authors have the option to publish the peer review history of their article (what does this mean?). If published, this will include your full peer review and any attached files.

Reviewer #2: No
